

# Single-particle experiments measuring humidity and inorganic salt effects on gas-particle partitioning of butenedial

Adam W. Birdsall[1], Jack C. Hensley[2], Paige S. Kotowitz[3*], Andrew J. Huisman[3**], Frank N. Keutsch[1,2,4]

[1]Department of Chemistry and Chemical Biology, Harvard University, Cambridge, MA, USA
[2]School of Engineering and Applied Sciences, Harvard University, Cambridge, MA, USA
[3]Chemistry Department, Union College, Schenectady, NY, USA
[4]Department of Earth and Planetary Sciences, Harvard University, Cambridge, MA, USA
*now at: nanoComposix, San Diego, CA, USA
**now at: Gentex Corporation, Zeeland, MI, USA

*Correspondence to*: Adam Birdsall (abirdsall@g.harvard.edu), Frank Keutsch (keutsch@seas.harvard.edu)

An improved understanding of the fate and properties of atmospheric aerosol particles requires a detailed process-level understanding of fundamental factors influencing the aerosol, including partitioning of aerosol components between the gas and particle phases. Laboratory experiments with levitated particles provide a way to study fundamental aerosol processes over timescales relevant to the multiday lifetime of atmospheric aerosol particles, in a controlled environment in
which various characteristics relevant to atmospheric aerosol can be prepared (e.g., high surface-to-volume ratio, highly concentrated or supersaturated solutions, changes to relative humidity). In this study, the four-carbon unsaturated compound butenedial, a dialdehyde produced by oxidation of aromatic compounds that undergoes hydration in the presence of water, was used as a model organic aerosol component to investigate different factors affecting gas–particle partitioning, including the role of lower-volatility "reservoir" species such as hydrates, time scales involved in equilibration between higher- and
lower-volatility forms, and the effect of inorganic salts. The experimental approach was to use a laboratory system coupling particle levitation in an electrodynamic balance (EDB) with particle composition measurement via mass spectrometry (MS). In particular, by fitting measured evaporation rates to a kinetic model, the effective vapor pressure was determined for butenedial and compared under different experimental conditions, including as a function of ambient relative humidity and presence of high concentrations of inorganic salts. Even under dry (RH<5%) conditions, the evaporation rate of butenedial is
orders of magnitude lower than what would be expected if butenedial existed purely as a dialdehyde in the particle, implying an equilibrium strongly favoring hydrated forms and the strong preference of certain dialdehyde compounds to remain in a hydrated form even under lower water content conditions. Butenedial exhibits a salting-out effect in the presence of sodium chloride and sodium sulfate, in contrast to glyoxal. The outcomes of these experiments are also helpful in guiding the design of future EDB-MS experiments.



## 1 Introduction

Atmospheric aerosol particles contribute significantly to health and climate effects (Boucher et al., 2013; Cohen et al., 2017). In order to understand and predict the extent and composition of aerosol particles across all environments, models of atmospheric aerosol require well-constrained experimentally derived parameters describing aerosol behavior over their
entire multiday lifetime in the atmosphere.

One key process is gas–particle partitioning, meaning a quantitative measure of what fraction of a given atmospheric chemical species exists in the gas compared to the particle phase under a given set of atmospheric conditions. For example, accurate understanding of gas–particle partitioning is necessary to understand to what extent a given compound contributes to aerosol particle loading in a particular environment. Despite seemingly being a straightforward question to
answer, reconciling observations of gas–particle partitioning with scientists' best understanding of the involved compounds remains an area of active research (Bilde et al., 2015).

The vapor pressure of a compound is an important parameter affecting its gas–particle partitioning behavior, but is not necessarily sufficient to completely describe the partitioning. Strictly speaking, a vapor pressure describes the equilibrium between pure condensed- and gas-phase forms of a particular compound. However in the complex matrix of
atmospheric aerosol particles, a compound can instead be thought of as exhibiting an "effective vapor pressure", $P_{vap,eff}$, meaning the vapor pressure apparently demonstrated by the compound when at equilibrium in a system consisting of a gas phase and single condensed phase, if the role of effects beyond Raoult's Law (i.e., a mole fraction dependence) were ignored. Similarly, while the Henry's law constant, $K_H$, is used to describe gas–particle partitioning of a compound in a dilute aqueous solution, the behavior of a compound in atmospheric aerosol can be described using an "effective Henry's
law constant", $K_{H,eff}$.

Researchers have used global-scale chemical transport models to investigate the effect of representation of $K_{H,eff}$ on modeled secondary organic aerosol (SOA) concentrations. In particular, $K_{H,eff}$ has been used as a parameter describing the loss of gas-phase oxidized semivolatile organic compounds (SVOC) via wet deposition, as a pathway competing with SOA formation. In these modeling studies, one approach has been to use a single uniform $K_{H,eff}$ of $10^3$ to $10^5$ M atm$^{-1}$ for SVOC
for which $K_{H,eff}$ is otherwise unknown (e.g., Pye and Seinfeld, 2010). An alternate approach has been to obtain a model-derived parameterized dependence of $K_{H,eff}$ on volatility that distinguishes between anthropogenic and biogenic precursors (Hodzic et al., 2014). Overall, calculated secondary organic aerosol loadings have been shown to depend on the representation of $K_{H,eff}$ (Hodzic et al., 2016; Knote et al., 2015), underlining the importance of an improved fundamental understanding of how $K_{H,eff}$ depends on the chemical behavior of a compound within the complex matrix of atmospheric
aerosol particles.

To this end, this research article focuses on two possible effects on $P_{vap,eff}$ and $K_{H,eff}$: formation of condensed phase "reservoir" species and nonideality in the condensed phase due to interactions with inorganic ions (Fig. 1).





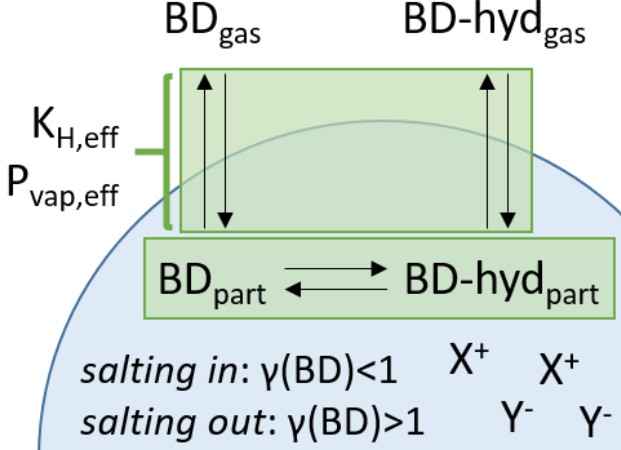

**Figure 1**: Schematic illustrating the meaning of effective vapor pressure, $P_{vap,eff}$, and effective Henry's law constant, $K_{H,eff}$. A compound partitioning between the gas and particle phase (here, butenedial, BD) can exist in the gas phase ($BD_{gas}$). In the particle phase the compound can exist in its standard form ($BD_{part}$) but also in fast equilibrium with a "reservoir" species via reversible chemical equilibrium (here, hydrated butenedial, BD-hyd$_{part}$). The reservoir species also can partition to the gas phase (BD-hyd$_{gas}$), but with a lower vapor pressure. The formal vapor pressure and Henry's law constant of butenedial are related to the equilbrium between $BD_{gas}$ and $BD_{part}$ alone. In contrast, $P_{vap,eff}$ and $K_{H,eff}$ are related to the equilibrium between the gas phase and the combined particle-phase abundance of both BD and reservoir species BD-hyd. Additionally, $P_{vap,eff}$ and $K_{H,eff}$ can be modulated by the presence of inorganic ions in the particle phase, shown here as generic cations $X^+$ and generic anions $Y^-$, which may have a salting-in or salting-out effect.

First, the compound may experience a second equilibrium in the condensed phase with a "reservoir" species with a lower vapor pressure. This reservoir species may be formed by chemical reactions such as hydration or oligomerization. If a large proportion of the compound exists in the form of the reservoir species, the result would be a lower $P_{vap,eff}$ or higher $K_{H,eff}$, whose value can depend on both the gas–particle partitioning equilibrium as well as on the equilibrium with the reservoir species (Ervens and Volkamer, 2010). The prototypical molecule for this process is the two-carbon dialdehyde glyoxal, which is known to exist largely in hydrated form in the presence of condensed-phase water, leading to a large $K_{H,eff}$ (Ip et al., 2009). The process here of reversible formation of a reservoir species is in contrast to the nonreversible formation of a product that takes place in the process of reactive uptake, for example by the isoprene epoxydiols (Lin et al., 2012).

A second effect could be if nonideal mixing, within a single phase, between the compound of interest and other aerosol components causes the activity coefficient of the compound, $\gamma$, to differ strongly from unity. Here $P_{vap,eff}$ or $K_{H,eff}$ differs from the value in a corresponding ideal mixture by a factor of $\gamma$. When the nonideality is due to interactions with inorganic compounds, which can be highly concentrated in ambient aerosol particles, the effect is called a "salting-in" ($\gamma<1$) or "salting-out" ($\gamma>1$) interaction, depending on whether the interaction favors or disfavors, respectively, the presence of the



compound in the phase in question. The effect of salting in or salting out on chemical systems with atmospheric relevance has been assessed with both experimental and theoretical approaches (Toivola et al., 2017; Wang et al., 2014; Waxman et al., 2015; Yu et al., 2011).

The gas–particle partitioning behavior of a given compound will depend on its chemical properties. In the present work we have studied the four-carbon unsaturated dialdehyde butenedial. Butenedial has been observed in the atmosphere and can be produced as a first-generation oxidation product of aromatic compound precursors (Birdsall and Elrod, 2011; Dumdei and O'Brien, 1984; Shepson et al., 1984). Furthermore we are interested in butenedial in terms of what its behavior reveals about the behavior of aldehydic compounds present in the atmosphere. One characteristic of butenedial is that the aldehyde functional groups are expected to readily hydrate under aqueous conditions, such as in an aqueous aerosol phase

whose water content is governed by the relative humidity (RH) of the surrounding gas phase (Fig. 2). The properties of butenedial, such as $P_{\mathrm{vap,eff}}$ or $K_{\mathrm{H,eff}}$, are expected to vary greatly depending on whether butenedial primarily exists in a hydrated or non-hydrated form, as has been observed for glyoxal. Another dicarbonyl compound, the three-carbon compound methylglyoxal, has also received appreciable study (e.g., Curry et al., 2018), though one notable structural difference is it contains one aldehydic group and one ketone. Unlike for glyoxal and methylglyoxal, $P_{\mathrm{vap,eff}}$ or $K_{\mathrm{H,eff}}$ for

butenedial previously has not been measured experimentally, to the best of our knowledge. Measurement of the gas–particle partitioning of butenedial provides further insight into the behavior of dicarbonyl compounds larger than glyoxal and methylglyoxal.

**Figure 2**: Butenedial (left) in equilibrium with its singly (center) and doubly (right) hydrated forms. The cis isomers of butenedial and its hydrates are shown here but no assumptions are made about the geometric isomer studied in this work.

One approach to address these questions is via laboratory experiments studying individual particles levitated in a chamber surrounded by a bath gas. The advantages of this approach include the ability to isolate and quantify individual

processes of interest under controlled conditions. An additional strength of levitated particle experiments is the ability to create a system that exhibits certain key characteristics of atmospheric aerosol: a relatively high surface-area-to-volume ratio enabling investigation of the interaction between bulk and multiphase processes, a condensed phase that can achieve highly concentrated or supersaturated conditions that are often present in atmospheric aerosol particles, and the ability to levitate particles over time scales consistent with their multiday atmospheric lifetimes so that "slow" processes can be studied.

Furthermore, the ability to flow a pure bath gas continually through the levitation chamber simplifies the system compared to



analogous experiments performed in an environmental chamber, removing the need for continual gas-phase measurements and reducing the set of multiphase processes that need to be accounted for, such as the effects of a more complex gas-phase matrix. The observations made in levitated particle experiments can, with care, be extrapolated to ambient atmospheric conditions, for example, by accounting for the difference in size (and hence surface-area-to-volume ratio) between laboratory and atmospheric particles.

Previous work in our laboratory has developed a technique termed electrodynamic balance–mass spectrometry (EDB-MS) to levitate individual charged droplets with diameter on the order of 10 μm in an electrodynamic balance (EDB) and then measure the droplet's composition with mass spectrometry (MS) (Birdsall et al., 2018). Other groups have developed methods to measure vapor pressures of organic compounds using highly precise particle diameter measurements using optical sizing techniques (Krieger et al., 2012). One difference between MS and optical sizing approaches is with MS, vapor pressures of individual chemical components can be readily extracted in a multicomponent system. In this research article, we report experiments in which we used our EDB-MS instrumentation to measure $P_{vap,eff}$ of butenedial under different experimental conditions, by measuring butenedial's evaporation rate and fitting the data to a kinetic model in which $P_{vap,eff}$ is the free parameter. $K_{H,eff}$ can also be calculated based on a calculated particle water content, using measured relative humidity and a thermodynamic model. First we measured $P_{vap,eff}$ of butenedial in purely organic aerosol particles under two relative humidity conditions. We then measured $P_{vap,eff}$ of butenedial in mixed organic/inorganic aerosol particles, containing either sodium chloride or sodium sulfate under humid conditions. The results demonstrate the possible role of different factors on the effective vapor pressures and Henry's law constants of organic aerosol components, as well as help inform future EDB-MS experiments.

## 2 Experimental

### 2.1 Butenedial synthesis

Butenedial was synthesized in our laboratory following literature procedure (Avenati and Vogel, 1982). In brief, 2.4 M 2,5-dihydro-2,5-dimethoxyfuran (DMDF, TCI America, 98%) was hydrolyzed at 25° C over ~10 d in aqueous solution containing acetic acid (VWR, 99.7%) diluted to 3.4 M in deionized water. Reaction progress was monitored via NMR. Rotary evaporation was used to remove acetic acid, residual DMDF, and excess water. The mass fraction of the rotovapped solution determined to be butenedial was quantified via quantitative addition of diethylmalonic acid (Sigma Aldrich, 98%) as an internal standard, and found to be 0.75±0.02, with the lack of major unidentified peaks in the spectrum implying the balance of the composition to consist of water. No further purification was deemed necessary based on the collected product NMR spectrum (Fig. 3).





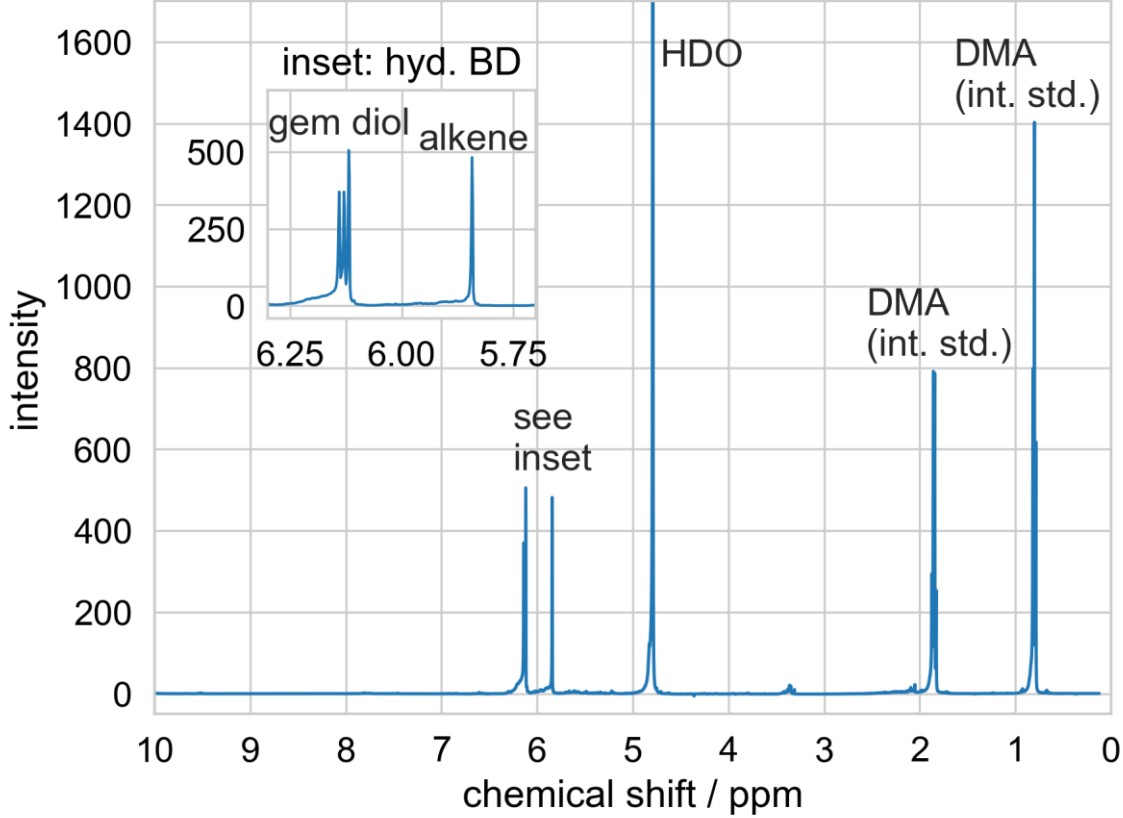

**Figure 3**: NMR spectrum of synthesized butenedial in deuterated water solvent. Includes peaks associated with hydrated butenedial (hyd. BD), diethylmalonic acid (DMA), used as an internal standard (int. std.), and solvent (HDO).

**2.2 Droplet levitation and measurement**

Microdroplets were generated, levitated, and measured via mass spectrometry using the same electrodynamic balance–mass spectrometry (EDB-MS) apparatus as previously described (Birdsall et al., 2018). In brief, a droplet-on-demand particle generator injects a ~140 pL droplet from an aqueous solution into the EDB levitation chamber, through a charged coil that gives the droplet charge. A combination of AC and DC electrodes, in a "dual-ring" geometry, establishes an
10 electric field that confines the charged droplet in the center of the levitation chamber. The water content from the aqueous solution rapidly equilibrates with the relative humidity of the levitation chamber (~1 s). A single droplet is allowed to reside in the center of the levitation chamber for a defined period of time while a purge flow of dry or humidified nitrogen gas flows through the levitation chamber (95 standard cubic centimeters per minute, sccm), maintaining a fixed relative humidity (RH) in the levitation chamber and preventing the accumulation of gases evaporating from the droplet. At the end of the
15 residence time in the droplet levitation chamber, the gas flow and electric field are manipulated to eject the droplet out of the



bottom of the levitation chamber, through a straight length of 1/4" (outer diameter) stainless steel tubing to the ionization region. In the ionization region, which like the levitation chamber is at ambient pressure, the droplet impacts a glass slide mounted on a heated plate (220° C) and the resulting vapors are ionized via a corona discharge emanating from a charged needle and drawn into the inlet of a commercial time-of-flight mass spectrometer, operated at unit mass resolution (JEOL

5    AccuTOF). A sample mass spectrum of a single particle containing butenedial, hexaethylene glycol (PEG-6, an internal standard) and sodium chloride is shown in Fig. 4.

**Figure 4**: Sample mass spectrum of droplet trapped in EDB and measured with MS, using the BD + PEG-6 + NaCl ($X_{NaCl}$ =

10    0.140) experimental solution. Both the raw and background-subtracted spectra are shown. Butenedial is observed at its parent ion of 85 m/z. PEG-6 is observed at its parent ion of 283 m/z, along with at known fragment m/z labeled with *.

Mass spectra were collected using commercial software (JEOL MassCenter), resulting in a brief (few second) signal "pulse" above the background on m/z channels corresponding to components of the vaporized droplet (Fig. 5). As





previously, the intensity at each m/z channel was quantified as the height of the pulse above the background. The peak height calculation was automated using an algorithm that detected the peak at each mass channel of interest and subtracted an average background value from before the peak onset, and was checked by eye for correctness.

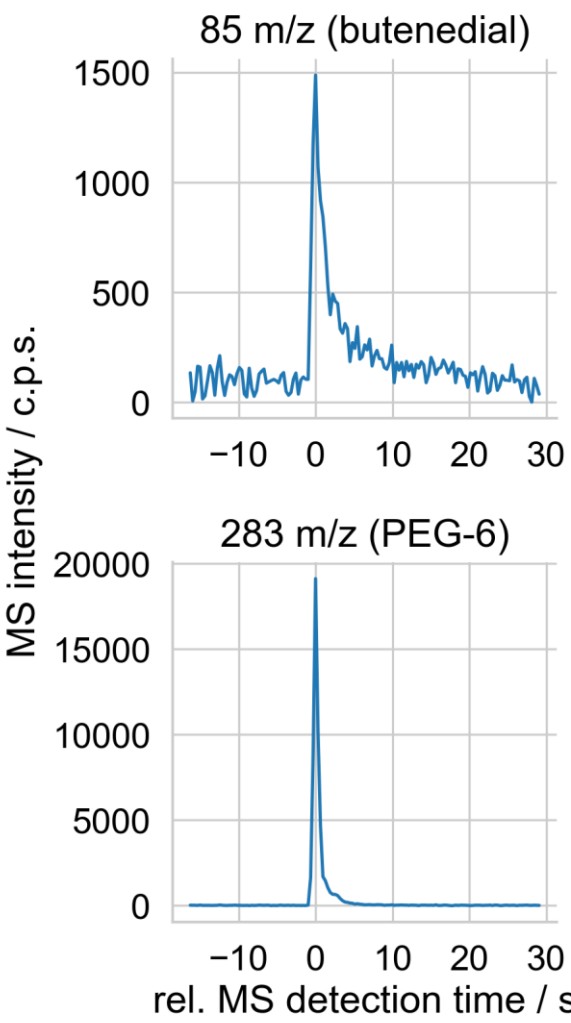

**Figure 5**: Sample extracted ion time series from measurement of a single droplet, whose mass spectrum is shown in Fig. 4. The peak intensities in the 85 and 283 m/z mass channels above the surrounding background are used to quantify the abundance of butenedial and PEG-6 in the droplet, respectively.

10       One change from the previously described setup was to sample the mass spectrometer at a faster rate of 3 Hz, compared to previous sampling at 1 Hz. Each mass spectrum had sufficient signal, with better temporal sampling of the droplet signal peak. With the current set of experiments, we observed less particle-to-particle variability in the normalized





analyte signal compared to previously (Birdsall et al., 2018). This could potentially be explained by our previous technique undersampling the peak shape.

As previously described (Birdsall et al., 2018), droplet sizes were characterized after initial trapping and before the ejection procedure using a spring point technique, which relies on a quantitative relationship of the visible onset of droplet

instability (the "spring point") in the EDB with the voltage conditions and the droplet diameter (Davis, 1985). The droplet diameters determined for this study used the same calibration data using 18 µm diameter polymethylmethacrylate (PMML) spheres as previously. The absolute accuracy of this technique is estimated at 20%.

### 2.3 Experimental systems

We used two types of experiments to probe the effect of relative humidity (RH) and inorganic content on

evaporation rate of butenedial and on extracted values of $P_{\mathrm{vap,eff}}$ and $K_{\mathrm{H,eff}}$. The experimental solutions used in the current study are reported in Table 1. Overall two types of experiments were performed.

**Table 1**: Parameters describing the experimental conditions used in each experiment type. These parameters and associated uncertainties were used as model inputs. Relative molar quantities of butenedial, PEG-6 and the inorganic salts were

determined via the mass composition of the prepared precursor solution. The overall mole fraction of water was calculated from the experimental relative humidity (75±5% for the humid experiments and 2.5±2.5% for the dry experiments) and the solution composition via the thermodynamic model AIOMFAC (see body text). Radius measurements were obtained via a spring point technique (see body text). Provided uncertainties are 1σ values and incorporated into the model calculations as explained in the main text. *Radius distribution imputed from dry BD + PEG-6, adjusted by factor of 1.12 to account for

presence of water. **Radius distribution imputed from NaCl (#1) experiment.

| composition | RH | $X_{\mathrm{BD}}$ | $X_{\mathrm{PEG-6}}$ | $X_{\mathrm{NaCl}}$ | $X_{\mathrm{Na_2SO_4}}$ | $X_{\mathrm{H_2O}}$ | $r$ (µm) | $T$ (K) | $N$ |
|---|---|---|---|---|---|---|---|---|---|
| BD + PEG-6 | <5% | 0.457 ± 0.013 | 0.518 | 0 | 0 | 0.025 ± 0.013 | 12.9 ± 1.5 | 300.10 ± 0.75 | 19 |
| BD + PEG-6 | 75 ± 5% | 0.0960 ± 0.0028 | 0.109 | 0 | 0 | 0.795 ± 0.013 | 14.4 ± 1.7* | 300.10 ± 0.75 | 9 |
| BD + PEG-6 + NaCl (#1) | 75 ± 5% | 0.0976 ± 0.0026 | 0.0514 | 0.071 | 0 | 0.780 ± 0.020 | 13.8 ± 2.1** | 300.10 ± 0.75 | 16 |
| BD + PEG-6 + NaCl (#2) | 75 ± 5% | 0.0255 ± 0.0007 | 0.0145 | 0.140 | 0 | 0.820 ± 0.015 | 13.8 ± 2.1** | 300.10 ± 0.75 | 17 |
| BD + PEG-6 + Na₂SO₄ | 75 ± 5% | 0.0621 ± 0.0027 | 0.0528 | 0 | 0.105 | 0.795 ± 0.018 | 13.8 ± 2.1** | 300.10 ± 0.75 | 8 |



One type of experiment was designed to investigate the effect of RH on the evaporation rate (and hence $P_{vap,eff}$) of butenedial. These experiments used droplets from a solution containing a mixture of butenedial and hexaethylene glycol (PEG-6, i.e., the polyethylene glycol hexamer, 99%, Sigma Aldrich), the latter of which served as an internal standard for reasons discussed in Sect. 2.3.1. Droplets were exposed to two different RH levels: "dry" experiments for which the bath gas

was dry nitrogen gas with RH in the EDB measured to be <5%, and "humid" experiments for which the bath gas was humidified nitrogen gas created by flowing nitrogen gas through a water bubbler with RH in the EDB measured to be 75 ± 5%.

A second type of experiment was designed to probe the effect of the presence of inorganic species on $P_{vap,eff}$ of butenedial. As for the RH experiments, droplets contained butenedial along with PEG-6 acting as an internal standard.

Additionally the droplets contained one of two inorganic compounds, either sodium chloride (NaCl) or sodium sulfate (Na$_2$SO$_4$). Two NaCl solutions were studied with $X_{NaCl}$ of 0.071 and 0.140; one Na$_2$SO$_4$ solution was studied with $X_{Na_2SO_4}$ of 0.105. These experiments were performed with a humidified levitation chamber at RH 75 ± 5%.

Separate trials in a similarly designed EDB equipped with additional spectroscopic instrumentation (described in Sect. 3.2 of Krieger et al. (2018)) were performed to confirm the particles remained deliquesced at our experimental

humidity for the inorganic particles. The observed two-dimensional angular optical scattering pattern remained visible at the experimental humidity, implying a spherical deliquesced particle (Braun and Krieger, 2001). No change in fringe pattern was observed when drying a particle containing butenedial and PEG-6 without an inorganic salt, meaning that no phase separation or efflorescence occurred in the salt-free system. In contrast, when dry nitrogen was introduced as the bath gas for salt-containing particles, the fringe pattern was observed to change, indicating that efflorescence had occurred due to the

presence of the inorganic salt.

### 2.3.1 PEG-6 as internal standard

We deemed PEG-6 to be an appropriate internal standard and major condensed-phase component for these experiments for a number of reasons. PEG-6 was used as an internal standard to account for droplet-to-droplet variability in total amount of signal measured with the mass spectrometer because it shows a clear peak at 283 m/z in the mass spectrum

(MH$^+$) and its vapor pressure is low enough for its evaporation to be negligible over the timescale of the experiments. We also used PEG-6 as an internal standard in a previous set of experiments (Birdsall et al., 2018). Based on those previous results, the viscosity of a PEG-6 matrix is sufficiently low to expect no condensed-phase diffusion effects limiting the evaporation rate of butenedial.

We also verified our observations were not affected by reactive chemistry between PEG-6 and butenedial. Because

butenedial contains aldehyde groups and PEG-6 contains alcohol groups, conceivably a hemiacetal could be formed by their reaction, which would lower $P_{vap,eff}$ of butenedial. We repeated the humid evaporation measurements with diethylmalonic acid as the internal standard rather than PEG-6, for which no chemical reaction with butenedial is expected. We observed no





difference in evaporation rate, within experimental uncertainty, of butenedial between the PEG-6 and diethylmalonic acid data, implying the presence of PEG-6 did not measurably influence the evaporation rate. Furthermore, we observed no peaks in the mass spectrum at m/z ratios consistent with a PEG-6–butenedial hemiacetal. Recent work has shown that a model hemiacetal oligomer was detected intact via mass spectrometry when quickly (millisecond timescale) vaporized from impact

onto a heated rod at ~160 °C (Claflin and Ziemann, 2019), which suggests under our similar analytical conditions we would also expect to observe the intact hemiacetal, if present. Nor did we observe a peak in a 2D NMR heteronuclear multiple bond correlation (HMBC) experiment of the precursor PEG-6–butenedial aqueous solution consistent with coupling between the butenedial and PEG-6 moieties of a putative hemiacetal, as would be expected.

To check whether salting out of PEG-6 would affect the extracted butenedial $P_{\text{vap,eff}}$ in the inorganic experiments,

additional evaporation model runs (see Sect. 2.4) were performed in which the model representation of PEG-6 had a vapor pressure 10 times larger than the literature value. This was used as a conservative upper bound of a salting-out effect. With this higher vapor pressure of PEG-6, evaporation of PEG-6 still proceeded slowly enough over experimental timescales such that there was no discernible difference in the extracted butenedial $P_{\text{vap,eff}}$ (<1% change).

## 2.4 Evaporation model and determination of $P_{\text{vap,eff}}$ and $K_{\text{H,eff}}$

$P_{\text{vap,eff}}$ of butenedial under each experimental condition was determined by fitting a kinetic model that describes the changing composition of a droplet in time to observations. To determine the uncertainty in the extracted $P_{\text{vap,eff}}$, the analysis considered the uncertainty in the model input parameters and the standard error in the non-linear curve fitting coefficient, as detailed below. The data consists of a set of individual observations for each trapped and measured droplet, corresponding to a normalized abundance of remaining butenedial (relative to PEG-6) after butenedial evaporation has proceeded for a certain

amount of time. Plotting a set of these data points for a single type of experiments shows a decay over time in the normalized butenedial signal, which is compared to the kinetic model.

### 2.4.1 Maxwell flux description of particle evaporation

The kinetic model describing evaporation was implemented in pyvap, an open-source Python package that has been previously described (Birdsall et al., 2018). In brief, the kinetic model numerically integrates a differential equation

describing evaporation of droplet components via Maxwellian flux (Seinfeld and Pandis, 2006), as in Eq. 1:

$$\frac{dn_i}{dt} = 4\pi r D_{\text{g}_i}(c_{\infty_i} - c_{\text{s}_i}) \qquad (1)$$

in which $r$ is the particle radius, $D_{\text{g}_i}$ is the gas-phase diffusion constant of species $i$, and $c_{\infty_i}$ is the gas-phase concentration of

species $i$ at infinite distance from the particle surface (taken to be zero in the EDB), and $c_{\text{s}_i}$ the gas-phase surface



concentration of species $i$. We assume the particle is an ideal mixture and in equilibrium at its surface with the gas phase, so the gas-phase surface concentration is then given by Eq. 2:

$$c_{s_i} = X_i \frac{P_{\mathrm{vap}_i,\mathrm{eff}}}{kT} \qquad (2)$$

where $X_i$ is the particle-phase mole fraction of species $i$, $P_{\mathrm{vap}_i,\mathrm{eff}}$ is the pure component vapor pressure of species $i$ at temperature $T$ inside the EDB, and $k$ is the Boltzmann constant.

Due to the fast equilibration of water, its evaporation is not explicitly represented. Instead, the water content is calculated at each time step to preserve a fixed mole fraction of water in the particle. The evaporation of PEG-6 is explicitly

represented using experimental physical parameters available in a review of polyethylene glycol vapor pressures (Krieger et al., 2018). However, over the experimental timescales involved the evaporation of PEG-6 was calculated to be negligible, making for our purposes PEG-6 effectively involatile. The model representations of sodium chloride and sodium sulfate had vapor pressures of 0, preventing any modeled evaporation. Sodium chloride and sodium sulfate are represented as being fully dissociated into two and three ions, respectively. The model does not include any representation of acid-base

equilibrium.

To make the fitting procedure less computationally expensive, rather than repeatedly numerically integrating the differential equation we developed an analytical expression that approximates a solution to Maxwellian flux under certain simplifying assumptions. In particular, we assumed droplet radius and mole fraction of water were time-invariant, and all non-butenedial, non-water droplet components did not evaporate over the experimental time scale. A description of the

resulting analytical solution and fitting procedure, along with validation of the technique, are given in Sect. S1 and S2.

### 2.4.2 Parameter inputs and uncertainty

All parameters aside from $P_{\mathrm{vap},\mathrm{eff}}$ of butenedial are constrained by knowledge of the experimental system. These parameters include the initial mole fraction of butenedial, temperature, RH, the gas-phase diffusivity of butenedial, and the initial droplet radius. The droplet diameter is provided by the spring-point measurements The temperature and RH is

provided by a RH/temperature probe (Sensirion SHT31).

Physical parameters used to represent each compound are given in Table 2. Other model input parameters are given in Table 1.



**Table 2**: Physical parameters for each compound used in the model. *Note that because all observations of butenedial in the condensed phase were consistent with existing in its dihydrated form, the model used physical parameters of butenedial dihydrate. **Because inorganic salts sodium chloride and sodium sulfate are essentially involatile, their vapor pressures were assumed to be effectively 0.

| compound | $M$ (kg / mol) | $\rho$ (kg / m³) | $P$(298K) (Pa) | $D_{gas}$ ($10^{-6}$ m² / s) |
|---|---|---|---|---|
| butenedial dihydrate | 0.1201* | 1060 | see results | $8.56 \pm 0.86$ |
| PEG-6 | 0.2823 | 1180 | $3.05 \times 10^{-5}$ | 4.26 |
| NaCl | 0.05844 | 2160 | 0** | n/a |
| Na$_2$SO$_4$ | 0.142 | 2660 | 0** | n/a |

The gas-phase diffusivity of butenedial (along with the dihydrate) in air was estimated using the Fuller-Schettler-Giddings equation, which is based on the molecular weights and volumes of the components of a binary gas mixture. The molecular volume of butenedial was calculated based on its molecular structure using literature parameters (Welty et al., 1984). The density of butenedial is assumed to be 1.06 g m$^{-3}$ based on literature densities of aldehydes butanedial and
glutaraldehyde (Lide, 2008).

The initial molar ratio of butenedial and other non-water components of the droplet (e.g., PEG-6, sodium chloride, sodium sulfate) is known from the relative concentrations of the compounds in the precursor aqueous solution, which was prepared quantitatively. However, the overall mole fraction of butenedial in the droplet depends on the droplet's water content. For the dry experiment the RH is measured to be 5% or lower. Though the possible presence of water in these low
amounts are accounted for in our uncertainty analysis (see below), we find under these conditions the presence of water has a negligible effect on accurately predicting the mole fraction of butenedial. In contrast for the humid experiments the water content is a non-negligible contribution to the overall molar composition. When equilibrated with the surrounding gas phase, the droplet water concentration will be such that the activity of water in solution and in the gas phase (i.e., RH) are equal.

To calculate the mole fraction of water corresponding to the RH of the humid experiments we used the
thermodynamic model AIOMFAC via its publicly accessible online interface (https://aiomfac.lab.mcgill.ca), using the known starting composition of the non-water particle components. AIOMFAC uses a functional group approach to calculate the equilibrium thermodynamic activities of a mixed inorganic/organic solution assumed to exist in a single deliquesced phase (Zuend et al., 2011, 2008). In particular, butenedial was defined in terms of its functional groups in its hydrated form because our results imply that is its dominant form in our experiments (see below). The mole fraction of water in the particle
was assumed to be fixed over the entire course of the experiment, assuming the activity coefficient of water does not change appreciably as butenedial evaporates from the particle.

Because of the different molar sensitivities of the mass spectrometer to butenedial and PEG-6, a scaling factor needed to be applied to scale the measured ratios of butenedial and PEG-6 signal intensities to the molar ratio. The scaling factor was determined independently for each type of experiment from the data points for the droplets that had resided in the




EDB for 5 minutes or less. Assuming negligible butenedial evaporation over this timescale, the molar ratio for these droplets was assumed to be equal to the known starting molar ratio of the precursor aqueous solutions. A bootstrapping procedure was used to estimate the mean and standard deviation in the short time interval normalized peak intensity, and from that along with the known molar composition of the starting precursor aqueous solution, a mean and standard deviation in the

scaling factor was determined.

Spring-point-derived radii were available for the dry experiment (16 radii, mean 12.9 μm and standard deviation 1.5 μm) and NaCl #1 experiment (16 radii, mean 13.8 μm and standard deviation 2.1 μm). Spring-point-derived sizes were not available for the humid (inorganic-free), NaCl #2, and $Na_2SO_4$ experiments. However, due to the similarity in droplet behavior (e.g., response to flow and voltage conditions during particle transfer) the distribution of droplet diameters for those

experiments could be imputed from the available measurements. The humid experiment diameter distribution was obtained by scaling the dry experiment diameter distribution by a factor of 1.115 to account for the calculated increase in diameter due to the presence of water. The NaCl #2 and $Na_2SO_4$ experiment diameter distributions were assumed to match that of the NaCl #1 experiment. The resulting diameter distributions used are given in Table 1.

We used a Monte Carlo approach to estimate the uncertainty in the retrieved butenedial $P_{vap,eff}$ due to uncertainties

in the other model input parameters. The implementation of the uncertainty analysis is described in Sect. S3. The analysis incorporates the uncertainty associated with model input parameters, including droplet diameter, temperature, scaling factor, and gas-phase diffusivity of butenedial, as reported in Table 1. Using this approach, the per-experiment uncertainties due to uncertainty in model input parameters and standard error in the model fit vary by experiment and are reported in Table 3, but range between ±20% and ±40%. Compared to other uncertainties in our approach, this is likely a dominant source of

uncertainty. For comparison, we previously described an uncertainty estimation technique in which we simply considered limiting cases of temperature and diameter parameters to give upper and lower bounds of $P_{vap,eff}$ of various polyethylene glycols (Birdsall et al., 2018). Using that less detailed uncertainty treatment, which did not consider uncertainty due to measurement variability, the previous estimated uncertainty range was between ±15% and ±25%.



**Table 3**: Extracted effective vapor pressures $P_{\text{vap,eff}}$ for butenedial (BD) with 1σ uncertainties obtained from uncertainties in model input parameters described in the main text, along with effective Henry's law constants $K_{\text{H,eff}}$ and Setschenow constants $K_S$, where applicable. Uncertainties in $K_{\text{H,eff}}$ and $K_S$ derived from propagating uncertainties in associated $P_{\text{vap,eff}}$ values.

| composition | RH | ionic strength (M) | $P_{\text{vap,eff}}(BD, 300K)$ (mPa) | $K_{\text{H,eff}}$ ($10^7$ M atm$^{-1}$) | $K_S$ (m$^{-1}$) |
|---|---|---|---|---|---|
| BD + PEG-6 | <5% | n/a | $31.1 \pm 9.0$ | n/a | n/a |
| BD + PEG-6 | $75 \pm 5\%$ | n/a | $39.5 \pm 8.6$ | $5.2 \pm 1.1$ | n/a |
| BD + PEG-6 + NaCl (#1) | $75 \pm 5\%$ | 5.3 | $58 \pm 18$ | $4.8 \pm 1.5$ | $+0.009 \pm 0.032$ |
| BD + PEG-6 + NaCl (#2) | $75 \pm 5\%$ | 9.6 | $224 \pm 89$ | $1.82 \pm 0.71$ | $+0.048 \pm 0.021$ |
| BD + PEG-6 + Na$_2$SO$_4$ | $75 \pm 5\%$ | 21.0 | $172 \pm 44$ | $1.54 \pm 0.40$ | $+0.073 \pm 0.020$ |

### 2.4.3 Effective Henry's law constant

Using the extracted $P_{\text{vap,eff}}$ along with the calculated molarity of butenedial, we estimated an effective Henry's law constant $K_{\text{H,eff}}$ for all humid experiments. The butenedial molarity was calculated using the water content calculated with the thermodynamic model AIOMFAC (Section 2.4.2) and assuming the overall molecular weight and density of the particle was

10 the weighted average of the molecular weights and densities of its chemical components, respectively. The full expression is then given by Eq. 3:

$$K_{\text{H,eff}} = \frac{X_{\text{BD}} MW_{\text{avg}}^{-1} \rho_{\text{avg}}}{X_{\text{BD}} P_{\text{vap,eff}}} \qquad (3)$$

or after simplification

$$K_{H,eff} = \frac{MW_{\text{avg}}^{-1} \rho_{\text{avg}}}{P_{\text{vap,eff}}} \qquad (4)$$

where $MW_{\text{avg}}$ is the weighted average of molecular weights of the chemical components of the particle and $\rho_{\text{avg}}$ is the

20 weighted average of densities of the chemical components of the particle.





Because the dominant source of uncertainty in Eq. 4 is $P_{\text{vap,eff}}$, we derive the uncertainty in $K_{\text{H,eff}}$ by using the same relative uncertainty as in the corresponding $P_{\text{vap,eff}}$.

## 3 Results and discussion

### 3.1 Humidity dependence

Figure 6 shows data and the corresponding best model fit for the humidity dependence butenedial evaporation experiments. The extracted values for the extracted $P_{\text{vap,eff}}$ of butenedial under the two RH conditions are $31.1 \pm 9.0$ mPa and $39.5 \pm 8.6$ mPa for the dry and humid conditions, respectively. (The uncertainty values correspond to a $1\sigma$ value derived from the Monte Carlo sampling approach described in the SI, and reflects the uncertainty due to uncertainty in the model input parameters and the standard error of the coefficient in the model fit.) We interpret these results to imply $P_{\text{vap,eff}}$ under

the dry and humid conditions are indistinguishable, within the uncertainties in our measurements and fitting procedure. Note the extracted $P_{\text{vap,eff}}$ values are similar despite the differences in evaporation timescales. This is a consequence of the evaporation rate scaling with the concentration of butenedial in the particle in Eq. 2. Because the mole fraction of butenedial decreases as water content increases, under humid conditions butenedial evaporation is expected to proceed more slowly even if $P_{\text{vap,eff}}$ is unchanged. This previously has been referred to as the "Raoult's Law effect" (Prisle et al., 2010).

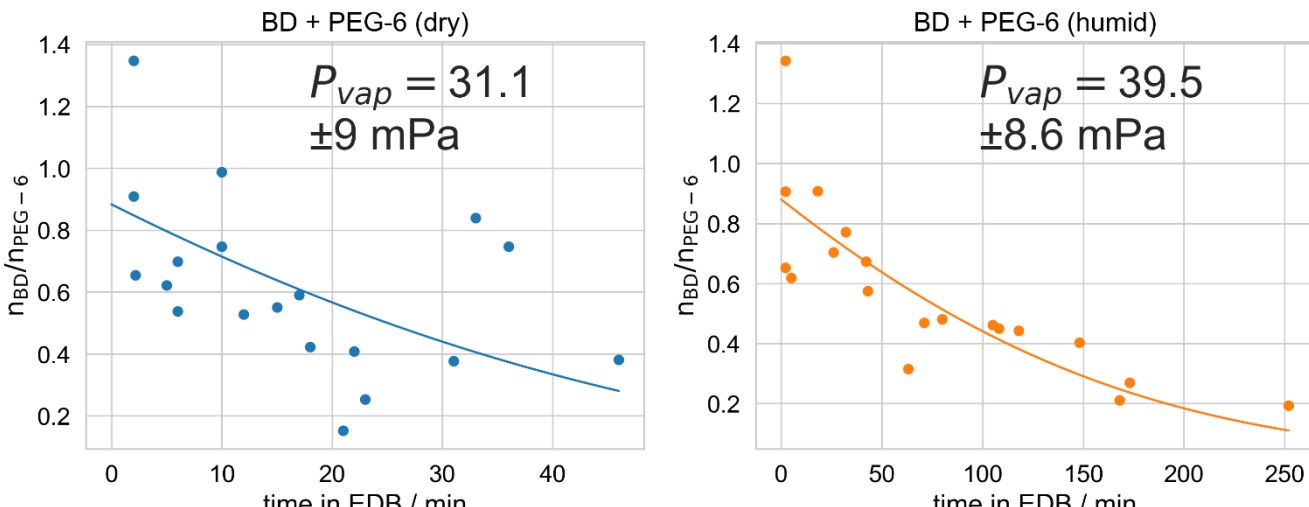

**Figure 6**: Experiments used to determine the effective vapor pressure of butenedial (BD) in a droplet also containing hexaethylene glycol (PEG-6), under dry (RH<5%) and humid (RH 75 ± 5%) conditions. Points are observations of individual droplets and the line is the best model fit, as described in the text, with associated vapor pressure and $1\sigma$

uncertainty printed.



One implication of the measured effective vapor pressures is that considering the hydration of butenedial is crucial for a correct prediction of its $P_{vap,eff}$ and hence its gas–particle partitioning behavior. If the butenedial in the particle primarily consisted of butenedial in its dialdehyde form, the expected $P_{vap,eff}$ would be expected to be orders of magnitude larger. For example, the EVAPORATION model (Compernolle et al., 2011) (accessed through UManSysProp (Topping et al., 2016)) predicts a vapor pressure for butenedial of 350 Pa at 298 K, and the Nannoolal et al. method (Nannoolal et al., 2008) (also accessed through UManSysProp) predicts an even higher vapor pressure of 1.5 kPa at 298 K. If butenedial were primarily in the form of the dialdehyde, we would expect to observe a much faster rate of evaporation than was actually observed. In contrast, using the dihydrated form of butenedial, EVAPORATION predicts a vapor pressure of 2.2 mPa at 298 K, which is almost within the same order of magnitude as the measured butenedial $P_{vap,eff}$. The Nannoolal et al. method predicts a very low vapor pressure of $2.5 \times 10^{-4}$ mPa at 298 K for the butenedial dihydrate.

Both EVAPORATION and the Nannoolal et al. method use a group contribution approach; the extremely low value of the Nannoolal et al. method prediction may reflect the limitations of its group contribution parameterization for this chemical structure. Additionally, in comparison with experimental vapor pressures, EVAPORATION has been reported to show a trend of underestimating vapor pressures at low vapor pressure (O'Meara et al., 2014). This trend could help account for the discrepancy between our measured $P_{vap,eff}$ and the EVAPORATION model estimate, though we cannot rule out the possibility that the discrepancy between measured $P_{vap,eff}$ and predicted butenedial dihydrate vapor pressure could possibly reflect a contribution to $P_{vap,eff}$ by the evaporation of butenedial monohydrate present in the particle. EVAPORATION predicts the monohydrate to have a vapor pressure of 2.3 Pa, whereas the Nannoolal et al. 2008 method predicts a vapor pressure of 220 mPa. However, the lack of any significant aldehyde peak in the NMR spectrum suggests equilibrium strongly favors the dihydrate form of butenedial under humid conditions.

The low observed $P_{vap,eff}$ of butenedial conceivably could also reflect a contribution from formation of butenedial oligomers, which would similarly lead to a lower $P_{vap,eff}$. However, we see no evidence for oligomer formation in our mass spectra or NMR spectrum. For oligomer formation to be consistent with our observations, it would have to be the case that ether oligomers hypothetically formed from butenedial completely decompose or fragment during the vaporization and ionization process into butenedial monomers that are detected at the butenedial ion parent ion value of 85 m/z. The likelihood of complete oligomer decomposition appears low. A recent study of different classes of model oligomers reported ether oligomers were detected intact via mass spectrometry after thermal desorption at ~160 °C (Claflin and Ziemann, 2019). Because we have no observational evidence for oligomer formation, in contrast to our observation of extensive butenedial hydration in NMR spectra, we conclude hydration of butenedial is the more likely explanation for the observation of a relatively low $P_{vap,eff}$. Finally, as elucidated in Sect. 2.3.1, we observe no evidence for formation of cross-products of butenedial reacting with PEG-6 that would reduce the $P_{vap,eff}$ of butenedial.



Furthermore the fact that the two $P_{\text{vap,eff}}$ are indistinguishable, at least within the estimated uncertainty of ~30%, implies butenedial primarily exists in a hydrated form not only under conditions with a high water content, but also under conditions with lower water content. This observation is also consistent with the collected NMR spectra of butenedial, in which only hydrated butenedial peaks are observed, even under conditions with a lower residual water content. There may be to some degree a shift in equilibrium between the hydrated and non-hydrated forms of butenedial under the different RH conditions, which could lead to a change in $P_{\text{vap,eff}}$, but our results imply any change would correspond to a change in $P_{\text{vap,eff}}$ less than our uncertainty in extracted $P_{\text{vap,eff}}$, approximately 20%.

A possible mechanistic explanation for this observation could be due to slow kinetics of dehydration under dry conditions. For all experiments, regardless of RH, the particle was initially generated from an aqueous solution in which equilibrium appears to strongly favor butenedial dihydrate. When the droplet is injected into the EDB under dry conditions, the water content of the particle quickly (~1 s) equilibrates. Within the dry particle, butenedial may be kinetically frozen over experimental timescales in the dihydrate form, even if the dialdehyde becomes more thermodynamically favorable. Additional experiments would be necessary to test this explanation.

Using Eq. 4 we calculated $K_{\text{H,eff}}$ of butenedial to be $5.2 \pm 1.1 \times 10^7$ M atm$^{-1}$ in the humid, inorganic-free experiment. (Uncertainties in $K_{\text{H,eff}}$ arise from propagating uncertainties in $P_{\text{vap,eff}}$, given the relationship in Eq. 4.) For comparison, $K_{\text{H,eff}}$ of glyoxal has been previously measured to be $4.19 \times 10^5$ M atm$^{-1}$ in an inorganic-free aqueous phase (Ip et al., 2009). The magnitude of the measured $K_{\text{H,eff}}$ for butenedial compared to glyoxal suggests butenedial may have a strong tendency to partition into an available aqueous phase, ignoring the effect of inorganic compounds.

**3.2 Inorganic salt dependence**

Figure 7 shows data for inorganic salt dependence experiments. For both the sodium chloride and sodium sulfate experiments, the extracted butenedial $P_{\text{vap,eff}}$ is larger than the $P_{\text{vap,eff}}$ measured in the organic-only cases, with values of 58 $\pm$ 18 mPa and 224 $\pm$ 89 mPa for the $X_{\text{NaCl}}$ of 0.071 and 0.140 sodium chloride experiments, respectively, and 172 $\pm$ 44 mPa for the sodium sulfate experiment. The fact that the $P_{\text{vap,eff}}$ for butenedial becomes higher in solutions containing both inorganic salts, by up to a factor of 8 under our experimental conditions, implies the inorganic salts in this case have a salting-out effect. Interestingly, previous work with a different dialdehyde, glyoxal, demonstrated that the presence of both sodium chloride and sodium sulfate led to a shift in the hydration equilibrium favoring the formation of the hydrate, implying a salting-in effect (Waxman et al., 2015; Yu et al., 2011). In contrast, the three-carbon dicarbonyl methylglyoxal has been measured to exhibit a salting-out effect for both sodium chloride and sodium sulfate (Waxman et al., 2015). The fact that salting-out effects have now been observed for both methylglyoxal and butenedial suggests the direction of salting in/salting out for a dicarbonyl compound may be influenced by the extent to which it contains hydrophobic regions, represented in butenedial by the alkenyl group and in methylglyoxal by the methyl group.





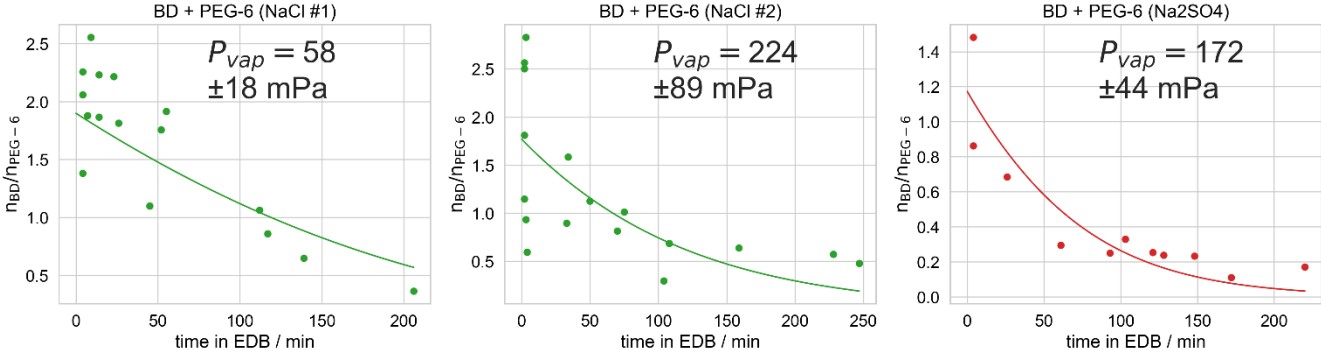

**Figure 7**: Experiments used to determine the effective vapor pressure of butenedial (BD) in a droplet also containing hexaethylene glycol (PEG-6) and either sodium chloride (NaCl) or sodium sulfate (Na₂SO₄), under humid (RH 75 ± 5%) conditions. Points are observations of individual droplets and the line is the best model fit, as described in the text, with associated vapor pressure and 1σ uncertainty printed.

Calculated $K_{H,eff}$ for the three inorganic salt-containing experiments, again using Eq. 4, are given in Table 3. Furthermore the magnitude of the salting-out effect can be quantified using the Setschenow coefficient $K_S$, using the formulation of Waxman et al. (2015) in Eq. 5, where

$$K_S = \frac{1}{c_s \log \frac{K_{H,w}}{K_{H,s}}} \qquad (5)$$

and $c_s$ is the condensed-phase salt concentration (using molality, m), $K_{H,w}$ is the effective Henry's law constant in the absence of the inorganic salt, and $K_{H,s}$ is the effective Henry's law constant in the presence of the inorganic salt. The calculated values of $K_S$ using Eq. 5 for the three inorganic experiments are given in Table 3, with uncertainties derived from propagating the uncertainties in the effective Henry's law constants. The $K_S$ values for the two NaCl experiments (+0.009 ± 0.032, +0.048 ± 0.021 m⁻¹) have 1σ uncertainty intervals that overlap. $K_S$ for sodium sulfate (+0.073 ± 0.020 m⁻¹) is approximately three times larger than the mean $K_S$ (+0.029 ± 0.019 m⁻¹) for sodium chloride in our experiments. Interestingly, this is a similar trend as previously reported for methylglyoxal in the presence of the same two inorganic salts, of 0.06 m⁻¹ and 0.16 m⁻¹, respectively (Waxman et al., 2015). However, the absolute magnitudes of the $K_S$ values are measured here to be smaller for butenedial compared to methylglyoxal. This may be due to butenedial having two aldehyde groups that are able to hydrate and have relatively energetically favorable interactions with the inorganic salts, compared to only one aldehyde group for methylglyoxal.

A competing hypothesis to potentially explain the measured higher $P_{vap,eff}$ in the mixed organic–inorganic droplets could be phase separation. If the presence of the inorganic components led to the formation of two different condensed



phases with butenedial predominantly in an organic-rich phase, the $P_{\text{vap,eff}}$ of butenedial would be higher due to its higher mole fraction concentration. However, thermodynamic calculations performed in AIOMFAC suggest this behavior is unlikely. Though a more extensive set of calculations would be necessary to rigorously check for the presence of partial phase separation, calculations performed using AIOMFAC assuming a single well-mixed phase under the experimental

conditions show the activities of the organic components remain well below 1 without extremely large (i.e., >>1) activity coefficients. These calculations are consistent with the droplets consisting of a single mixed phase under these experimental conditions.

### 3.3 Sources of uncertainty

The dominant source of uncertainty in our $P_{\text{vap,eff}}$ extraction procedure is uncertainty in the droplet diameter. The

uncertainty in the measured diameter for a population of particles is estimated to be up to a factor of 50% about the mean, which has a particularly large effect because vapor pressure scales with surface area. Using the spring point technique to measure the starting diameter of each trapped droplet has an inherent uncertainty. Additionally there is some degree in droplet-to-droplet variability in starting diameter for each population of droplets. Our modeling approach treats the diameter of a population of droplets for a single experiment type as a single mean value, with an associated standard deviation. In part

we treated the data this way because spring point-derived droplet diameters were not available for every droplet included in this analysis, though consistency in collected data across the data set implied each set of droplets could be described in terms of a single distribution of diameters. Additionally, previous analysis of a data set using spring point diameter measurements did not observe a correlation between residuals of the remaining analyte and the measured starting diameter for that particle, relative to the population mean (i.e., droplets measured using the spring point technique to be smaller than average did not

have their analyte molecule evaporate more quickly) (Birdsall et al., 2018). This suggests though there is considerable uncertainty in our spring point measurements, it would not be helpful to constrain the model based on the measured diameter of each individual droplet.

As a secondary effect, uncertainty in water content of the droplets as a calculated parameter may also contribute uncertainty to our $P_{\text{vap,eff}}$ determination. Our representation of the uncertainty in the mole fraction of water reflects solely

uncertainty in the relative humidity measurements upon which the model representation is based; we do not include a representation of the accuracy of the AIOMFAC model calculation used to determine the corresponding mole fraction of water itself. The water content is not used to constrain the droplet diameter (which is constrained by the spring point method, above), but it does have a significant effect on determining the overall mole fraction of butenedial in the droplet. For example, the mean measured RH for the humid experiments was 75%, but based on AIOMFAC model calculations the mean

mole fraction of water can be up to 0.82, depending on the hygroscopicity of the solution. Because the butenedial is a fixed mole fraction of the non-water portion of the solution, for the same set of measurements a difference in assumed mole fraction of water between 0.75 and 0.82 corresponds to a difference in extracted butenedial $P_{\text{vap,eff}}$ by a factor of (0.18-





0.25)/0.25, or almost 30%. Because the true uncertainty in the AIOMFAC activity correction is likely smaller than the difference between the ideal and calculated nonideal cases, 30% likely represents a highly conservative upper bound on uncertainty arising from the AIOMFAC water mole fraction calculation.

## 4 Conclusions

A set of laboratory experiments have used an electrodynamic balance–mass spectrometry (EDB-MS) technique to measure the evaporation of butenedial from organic-only and mixed organic–inorganic levitated droplets under dry and humid conditions. With this setup the specific process of interest, gas–particle partitioning, was studied in isolation in an environment in which a single particle of known composition is exposed to a continually refreshed bath gas of pure dry or humidified nitrogen. This approach simplifies the measurement compared to performing an experiment in an environmental

chamber, which will have a more complex gas-phase matrix, because no gas-phase measurement is required and no process other than evaporation needs to be modeled.

      We measured the effective vapor pressure ($P_{\text{vap,eff}}$) of butenedial, under both low (RH<5%) and higher (RH 70%) humidity conditions, to be approximately 30-40 mPa, which is 4 orders of magnitude lower than the expected vapor pressure of a four-carbon dialdehyde. This result implies butenedial exists primarily in a hydrated form, across a wide range of RH

conditions, and the gas–particle partitioning of butenedial in ambient particles favors the particle phase more strongly due to butenedial's hydration. The importance of hydration reactions in affecting gas–particle partitioning is consistent with previous work studying atmospherically relevant aldehydes, most notably glyoxal.

      These results emphasize the importance of considering the gas–particle partitioning of atmospheric compounds based on their actual chemical form in the condensed phase, rather than their pure form in isolation. For the case of

butenedial the current results suggest butenedial's $P_{\text{vap,eff}}$ can be represented by a single value across all ambient RH.

      The fact that $P_{\text{vap,eff}}$ is invariant between the two RH conditions may imply in these experiments butenedial remained in its hydrated form even under dry conditions with little condensed-phase water. This may be due to butenedial being kinetically frozen as a hydrate over the experimental timescales of tens of minutes to hours. If this is indeed the case, the ability of our EDB-MS approach to determine butenedial's $P_{\text{vap,eff}}$ is a consequence of butenedial hydrate evaporating

over a timescale during which the dehydration reaction is negligible. This analysis points to the care that needs to be taken when considering the behavior of atmospheric aerosol. If using an experimental approach with appreciably different conditions from those reported here (e.g., temperature or butenedial concentration), the rates of the competing evaporation and dehydration pathways may exhibit different scaling dependencies and hence lead to a different conclusion about butenedial's behavior. One attractive quality of EDB-MS is it allows monitoring the evolution of a particle's composition

over the multiday lifetime of an atmospheric aerosol particle. It is not necessary to use extreme experimental conditions to simulate some aspect of multiple days of atmospheric lifetime in a shorter period of time.



Observation of a higher $P_{vap,eff}$ of butenedial in the presence of high concentrations of inorganic salts sodium chloride or sodium sulfate imply a salting-out effect that increased $P_{vap,eff}$ of butenedial by up to a factor of approximately six to eight under the most concentrated inorganic conditions used in this experiment. In ambient particles the magnitude of the effect is expected to depend on the inorganic ion concentration, as described by the Setschenow coefficients. In general,

the magnitude of the observed Setschenow coefficients is such that only with inorganic ion concentrations on the scale of >1 M is a significant effect predicted. The salting-out effect is predicted to be negligible with inorganic ion concentrations on the order of 100 μM to 1 mM. Consequently, the measured salting-out effect is predicted to have an influence on aqueous aerosol particles but not on cloud water. The magnitude and sign of the salting-in or salting-out effect is also known to depend on the identity of the ion, so further study would be necessary to investigate the effect of other inorganic ions. In

general, better understanding of factors influencing gas–particle partitioning with this level of chemical detail improves predictions of the composition and fate of organic aerosol and its chemical constituents.

This work also helps inform the design of future experimental work using EDB-MS instrumentation. A measurement of particle diameter with lower uncertainty than the spring point technique would meaningfully reduce the uncertainty in extracted effective vapor pressures, particularly with a continuous diameter measurement. Other research with

EDBs has demonstrated the utility of optical sizing techniques for performing this measurement (Zardini et al., 2006).

In sum, this work uses an EDB-MS laboratory technique to better understand the dependence of a model organic compound's effective vapor pressure on particle composition, and hence help contribute to an improved understanding of fundamental factors influencing gas–particle partitioning in atmospheric aerosol.

**Data and code availability**

The Python package pyvap used as the kinetic model of droplet evaporation is available at https://github.com/awbirdsall/pyvap. Data used to generate paper figures is available upon request.

**Author contributions**

AWB, JCH, and FNK designed the experiments. AWB and JCH performed the laboratory experiments and analyzed the data. PK and AJH developed the electrodynamic balance used to assess droplet efflorescence behavior. JCH

developed the analytical solution approximating droplet evaporation and fitting procedure. AWB prepared the manuscript with contributions from all coauthors.

**Competing interests**

The authors declare they have no conflict of interest.



**Acknowledgments**

       This material is based upon work supported by NSF grant CHE 1808084, the National Science Foundation Graduate Research Fellowship under grant numbers DGE 1144152 and DGE 1745303, and the Harvard University Faculty of Arts and Sciences Dean's Competitive Fund for Promising Scholarship. The authors thank Ulrich Krieger for useful

5   discussions.



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
