# Peer review of "S1 Analytical solution to evaporation differential equation and curve-fitting procedure"

_Atmospheric Chemistry and Physics, 2019_

## Referee Comment (RC1) · Anonymous Referee #1 · 23 May 2019

This manuscript reports a study on the evaporation kinetics of butenedial using single particle levitation. The hypothesis is that the hydration of butenedial results in a low vapor pressure form that will evaporate more slowly than the non-hydrated form. This effect has been studied for other compounds (glyoxal, methylglyoxal etc.) but larger dicarbonyls, such as butenedial, have received little attention despite its observed presence in the atmosphere. Hydration will lead to much larger Henry's law uptake coefficients, with implications for gas-particle partitioning in the atmosphere.

This is particularly interesting and well-motivated research and the single particle MS technique reported has a lot of potential. However, I believe this is also one of the

flaws of the study – the technique itself is poorly suited to the measurements being made. That does not discredit the observations or invalidate the findings, but the large uncertainties are primarily a consequence of the experimental strategy, and similar methods could have produced data with clearer trends.

Overall, the paper is very well written and the work addresses the hypothesis, concluding that even under dry condition, the butenedial exists in the hydrated form. I recommend this for publication, although some minor points should be addressed as detailed below:

1) Please clarify the term "effective vapor pressure" – as far as I can tell, it is the vapor pressure that you'd calculate if you assume the activity coefficient to be unity. However, the description on Page 2 lines 15-20 is confusing.

2) Other single particle MS methods have been reported (Jacobs et al. 2018, for example) and should be cited.

3) The height of the pulse is used to quantify the abundance – what happens when you use the area of the pulse? Peak height is much more susceptible to peak shape effects (as evidence by the change in going from 1 Hz to 3 Hz) but peak area may be more robust.

4) What is the precision in the spring point method? Asked another way, how reliably can two droplets of a similar size be segregated by size?

5) The PEG is very hygroscopic and will drive the uptake of water at higher RH. It may also effectively hold on to water at low RH, possibly influencing the observations. Were measurements made with pure butenedial to verify if a faster rate of evaporation was observed under dry conditions?

6) The size of the particle is really key for these measurements given that only one component is evaporating. Typically, vapor pressure measurements are made from looking at the evolving size and then fitting these data to a model. It is unclear in

this study why MS was used as the sole measure of the amount of volatile material, when simple light scattering measurements would allow the size change to be resolved with high precision and accuracy. Clearly, for multi-component evaporation, the MS technique would be beneficial.

7) Figures 6 and 7 reveal a troubling amount of scatter in the measurement, demonstrating that even using an internal standard, a consistent measure of the composition is not possible. While vague trends are apparent, the uncertainty in the fit must be large. It is not clear if this is accounted for in the reported vapor pressure value. Please clarify. The MS response is not discussed as a source of uncertainty in 3.3, but clearly this is one of the more major sources. Droplets of different radii should exhibit the same relative intensity using the internal standard, so the variation seen in Figures 6 and 7 is an additional factor relating to the response of the instrument.

8) The evaporation rate is proportional to radius-squared, so an uncertainty of up to 50% could lead to an error in vapor pressure by a factor of 4.

9) I would like to see measurements reported for pure butenedial under dry and high RH conditions. Even if the data is crude due to lack of internal standard, this seems like a key measurement to aid in interpretation of the other data.

---

## Referee Comment (RC2) · Anonymous Referee #2 · 31 Jul 2019

I have little to add to the previous referee comment. I am worried about the unmeasured extent of water evaporation during the "dry" experiment. The panel of Figure 6 that reports on the MS result for the "dry" condition suggests a vapor pressure result (from the shown fit) that is much more precise than I believe, given the data. Scientifically, I wonder about the relationship between the empirical "salting out" that is documented here, with the availability of water for hydration of the aldehyde, given the high demand to solvate the inorganic ions. I suspect this is amenable to modeling, somehow.
* * *

---

## Author Response (AR1)

**Author Response for "Single-particle experiments measuring humidity and inorganic salt effects on gas–particle partitioning of butenedial" by A.W. Birdsall et al.**

We thank the referees for their thoughtful comments, which have helped improve the manuscript. Our replies are below (referee comment in **bold**, response in normal face, new manuscript content in *italics*, removed manuscript content in <del>strike-through</del>).

**Referee 1**

**1) Please clarify the term "effective vapor pressure" – as far as I can tell, it is the vapor pressure that you'd calculate if you assume the activity coefficient to be unity. However, the description on Page 2 lines 15-20 is confusing.**

We have updated the description of effective vapor pressure as follows (page 2, line 14):

[...] However in the complex matrix of atmospheric aerosol particles, a compound can instead be thought of as exhibiting an "effective vapor pressure",  $P_{vap,eff}$ , meaning the vapor pressure apparently demonstrated by the compound when at equilibrium in a system consisting of a gas phase and single condensed phase, if the role of effects beyond Raoult's Law (i.e., a mole fraction dependence) were ignored. In particular, the effective vapor pressure describes the vapor pressure that would be calculated if the activity coefficient were unity and condensed-phase chemical equilibria of the compound of interest with other "reservoir" forms were disregarded (see below). Similarly, while the Henry's law constant,  $K_{\rm H}$ , is used to describe gas-particle partitioning of a compound in a dilute aqueous solution, the behavior of a compound in atmospheric aerosol can be described using an "effective Henry's law constant",  $K_{\rm H,eff}$ .

**2) Other single particle MS methods have been reported (Jacobs et al. 2018, for example) and should be cited.**

The manuscript has been updated (page 5, line 6):

Previous work in our laboratory has developed a technique termed electrodynamic balance–mass spectrometry (EDB-MS) to levitate individual charged droplets with diameter on the order of 10  $\mu$ m in an electrodynamic balance (EDB) and then measure the droplet's composition with mass spectrometry (MS) (Birdsall et al., 2018). *Other single levitated particle MS methods have been reported (Jacobs et al., 2017).*

3) The height of the pulse is used to quantify the abundance – what happens when you use the area of the pulse? Peak height is much more susceptible to peak shape effects (as evidence by the change in going from 1 Hz to 3 Hz) but peak area may be more robust.

We have previously studied the effect of using peak area versus peak height to quantify the EDB-MS pulse signal. We found that, for systems of various polyethylene glycol oligomers similar to those described in a previous study (Birdsall et al., 2018), there was no appreciable difference in the normalized signal or shot-to-shot signal variability between the two quantification techniques.

**4) What is the precision in the spring point method? Asked another way, how reliably can two droplets of a similar size be segregated by size?**

Based on a previous study we estimate the precision in the spring point measurement to be ±10% (Birdsall et al., 2018). As discussed in the conclusions of the current manuscript (page 22, line 12), "A measurement of particle diameter with lower uncertainty than the spring point technique would meaningfully reduce the uncertainty in extracted effective vapor pressures, particularly with a continuous diameter measurement. Other research with EDBs has demonstrated the utility of optical sizing techniques for performing this measurement (Zardini et al., 2006)."

**5) The PEG is very hygroscopic and will drive the uptake of water at higher RH. It may also effectively hold on to water at low RH, possibly influencing the observations. Were measurements made with pure butenedial to verify if a faster rate of evaporation was observed under dry conditions?**

Because our experimental technique requires quantifying butenedial relative to an internal standard, we have found we are unable to collect meaningful data using pure butenedial without any internal standard. However, as noted in Sect. 2.3.1, we did perform experiments in which we measured butenedial evaporation using an alternate internal standard, the C7 dicarboxylic acid diethylmalonic acid. We observed no evidence of a faster butenedial evaporation rate in the presence of diethylmalonic acid.

6) The size of the particle is really key for these measurements given that only one component is evaporating. Typically, vapor pressure measurements are made from looking at the evolving size and then fitting these data to a model. It is unclear in this study why MS was used as the sole measure of the amount of volatile material, when simple light scattering measurements would allow the size change to be resolved with high precision and accuracy. Clearly, for multi-component evaporation, the MS technique would be beneficial.

We agree (and note in the manuscript) the uncertainty in the effective vapor pressures extracted from the set of experiments described in the current manuscript would be reduced with a more precise and accurate sizing measurement. This is capability we plan on adding to this instrumentation. However, using the MS technique we are able to quantify the evaporation rate sufficiently well to reach the scientifically interesting conclusions we describe. Furthermore, measuring the droplet using MS allowed us to check whether the droplet composition in fact remained solely PEG-6 and butenedial, or if other chemistry occurred (e.g., oligomerization). Using the MS technique for these experiments in which the condensed phase is a simple chemical system also will serve as a baseline for future studies based upon the same chemical system with additional condensed-phase chemical processes occurring.

7) Figures 6 and 7 reveal a troubling amount of scatter in the measurement, demonstrating that even using an internal standard, a consistent measure of the composition is not possible. While vague trends are apparent, the uncertainty in the fit must be large. It is not clear if this is accounted for in the reported vapor pressure value. Please clarify. The MS response is not discussed as a source of uncertainty in 3.3, but clearly this is one of the more major sources. Droplets of different radii should exhibit the same relative intensity using the internal standard, so the variation seen in Figures 6 and 7 is an additional factor relating to the response of the instrument.

Though we agree that there is a notable amount of scatter in the data, our results are consistent with the technique measuring a consistent measure of the composition, albeit with a significant source of shot-to-shot noise. In previous work we investigated potential sources of variability and found no systematic explanation (Birdsall et al., 2018). We have revised our fitting technique to better reflect the shot-to-shot variability. The uncertainty windows are now larger, as expected. There is also a somewhat wider confidence interval for the dry evaporation case than the humid, as appears should be the case from the relative noisiness of the data. The manuscript will be updated in the indicated sections with the following descriptions of the revised fitting procedure, along with updated vapor pressures uncertainty estimates obtained as a consequence of following this updated procedure:

Section 2.4 (page 11, line 14):

[revised manuscript text omitted]

|                                                 |            | strength | $P_{\rm vap, eff}(BD, 300K)$ | $K_{\rm H, eff}  (10^7)$ |                            |
|-------------------------------------------------|------------|----------|------------------------------|--------------------------|----------------------------|
| composition                                     | RH         | (M)      | (mPa)                        | M atm -1 )    | $K_{\rm S} ({\rm m}^{-1})$ |
| BD + PEG-6                                      | <5%        | n/a      | 28.1 (13.1, 47.8)            | n/a                      | n/a                        |
| BD + PEG-6                                      | 75 ±
5% | n/a      | 34.2 (18.8, 54.9)            | 6.0 (3.7,
11)         | n/a                        |
| BD + PEG-6 +
NaCl (#1)                       | 75 ±
5% | 5.3      | 66 (36, 105)                 | 3.1 (2.0,
5.7)        | +0.056
(0.012,
0.16) |
| BD + PEG-6 +
NaCl (#2)                       | 75 ±
5% | 9.6      | 169 (71, 301)                | 1.2 (0.68,
2.9)       | +0.074
(0.047,
0.15) |
| BD + PEG-6 +
Na 2 SO 4 | 75 ±
5% | 21.0     | 177 (64, 376)                | 1.2 (0.55,
3.2)       | +0.096
(0.056,
0.21) |

Section S3 (retitled *Combined* Monte Carlo *and bootstrapping* uncertainty analysis) (SI page 4, line 1):

The overall strategy of the *combined* Monte Carlo *and bootstrapping* uncertainty analysis was to obtain a distribution of extracted butenedial vapor pressures was obtained by repeating the fitting procedure in Sect. S1 10000 times, each time using a set of parameter values sampled at random from the set of distributions describing their uncertainties *and an independently generated bootstrapped realization of binned data*. The mean of the extracted butenedial vapor pressures provides a central value for the butenedial effective vapor pressure. The standard deviation describes the uncertainty due to uncertainties in the other model input parameters as well as *shot-to-shot noise in the data* <del>the standard error in the model fit coefficient</del>.

The source of the uncertainty in diameter arises from a combination of inherent uncertainty in the measurement and droplet-to-droplet variability, though the characteristics of each droplet were kept as consistent as possible. The uncertainties in gasphase diffusivity and scaling factor reflect uncertainties in the underlying parameters, rather than reflecting any variability in the values from particle to particle. The uncertainty in temperature does reflect the extent to which the EDB temperature drifted with time, though it should be noted the effect of temperature on the evaporation model over this range is limited. Each input parameter was represented by a Gaussian distribution centered at the mean value and with standard deviation based upon the variability or uncertainty in its measurements. The Monte Carlo approach assumes independence between each of the model input parameters, which is a reasonable assumption for this set of parameters. The distribution of each input parameter was treated separately for each experiment type (i.e., dry, humid, NaCl #1, NaCl #2, Na2SO4).

For each type of experiment, we binned the data into different time periods: those for which the time residing in the EDB was approximately 0 minutes, and then a series of equally spaced bins such that a total of 4 time bins were obtained. For each of the 10000 repetitions of the model fitting procedure, a bootstrapping procedure was used within each time bin to generated a bootstrapped realization of the normalized signal response. The model was fit to the mean value of the bootstrapped data within each time bin. The data was scaled for each trial assuming the bootstrapped mean for the "t=0" bin represents the initial normalized molar abundance of butenedial relative to the internal standard.

To calculate  $P_{\text{vap,eff}}$  for each iteration of the Monte Carlo technique, a value 95% confidence interval of *a* was estimated using the interval that *encompassed the extracted model fit for* 95% of the 10000 model fitting trials accounted for the standard error in the value of *a* extracted from the curve-fitting procedure. A single value of *a*-was sampled from a standard distribution centered at the optimal estimate of *a*, arising from the current iteration of curve-fitting procedure, and with standard deviation equal to the square root of the variance of the *a*-estimate, again from the current iteration of the curve-fitting procedure 95% confidence interval of *a* in Eq. S1, the value of *a* 95% confidence interval for  $P_{\text{vap}}$  was calculated for a single iteration of the Monte Carlo technique.

**8) The evaporation rate is proportional to radius-squared, so an uncertainty of up to 50% could lead to an error in vapor pressure by a factor of 4.**

As noted in the manuscript and responses above, we agree future experiments will be helped by additional accuracy in the size measurement.

**9) I would like to see measurements reported for pure butenedial under dry and high RH conditions. Even if the data is crude due to lack of internal standard, this seems like a key measurement to aid in interpretation of the other data.**

Please see our response to point 5, above. Unfortunately, these measurements with the current experimental setup provide no information.

**Referee 2**

I have little to add to the previous referee comment. I am worried about the unmeasured extent of water evaporation during the "dry" experiment. The panel of Figure 6 that reports on the MS result for the "dry" condition suggests a vapor pressure result (from the shown fit) that is much more precise than I believe, given the data. We have revised our fitting procedure to better reflect measurement uncertainties, please see response to point 7 of Referee 1.

Scientifically, I wonder about the relationship between the empirical "salting out" that is documented here, with the availability of water for hydration of the aldehyde, given the high demand to solvate the inorganic ions. I suspect this is amenable to modeling, somehow.

Thank you for the interesting suggestion. A future modeling study looking with more detail at the competition you describe between inorganic ion solvation and hydration of the aldehyde would provide helpful insights, but is beyond the scope of the current study.

**Co-editor comments**

Have you considered the possibility of intramolecular hydrogen bonds forming in the butadiene hydrates (Fig. 2) that would have a significant effect on their volatility and other chemical properties (perhaps also including their propensity to undergo condensation/oligomerization reactions)?

We agree hydrogen bonding may have important effects on volatility and other chemical properties. We will add the following paragraph to the manuscript in the Conclusions (at page 21, line 21):

The formation of intramolecular hydrogen bonds by hydrated butenedial may have a substantive effect on its volatility. Based on the structure of butenedial hydrate, the role of cis/trans isomerism is expected to play a role, with the cis form of butenedial dihydrate more likely to be able to form intramolecular hydrogen bonds and therefore demonstrate a high vapor pressure, compared to the trans form. Though the isomeric form of the precursor (cis) combined with the synthetic mechanism suggest synthesis of purely cis butenedial, evidence from our NMR spectra does not support this conclusion, instead suggesting our experiments were performed with a mixture of cis and trans isomers. However, we do not observe a double exponential shape to our evaporation data, which could imply either the vapor pressures of the cis or trans isomers do not have appreciably different vapor pressures, or the difference in evaporation rates is obscured by the noise in our data. Future studies on a wider set of compounds may help illuminate the effect.

There is a large body of other more recent work that can be discussed and cited regarding the accuracy in particle size that can be obtained from levitated particle trap techniques such as EDB and optical tweezers. The work of Jonathan Reid, Ulrich Krieger, and Ruth Signorell come to mind. A few suggestions:

Steimer, S. S.; Krieger, U. K.; Te, Y.-F.; Lienhard, D. M.; Huisman, A. J.; Luo, B. P.; Ammann, M.; Peter, T. Electrodynamic Balance Measurements of Thermodynamic, Kinetic, and Optical Aerosol Properties Inaccessible to Bulk Methods. Atmos. Meas. Tech. 2015, 8 (6), 2397–2408. Marsh, A.; Rovelli, G.; Song, Y.-C.; Pereira, K. L.; Willoughby, R. E.; Bzdek, B. R.; Hamilton, J. F.; Orr-Ewing, A. J.; Topping, D. O.; Reid, J. P. Accurate Representations of the Physicochemical Properties of Atmospheric Aerosols: When Are Laboratory Measurements of Value? Faraday Discuss. 2017, 200, 639–661.

Haddrell, A. E.; Davies, J. F.; Reid, J. P. Time-Resolved Measurements of the Evaporation of Volatile Components from Single Aerosol Droplets. Aerosol Science and Technology. 2012, pp 666–677.

Hargreaves, G.; Kwamena, N.-O. A.; Zhang, Y. H.; Butler, J. R.; Rushworth, S.; Clegg, S. L.; Reid, J. P. Measurements of the Equilibrium Size of Supersaturated Aqueous Sodium Chloride Droplets at Low Relative Humidity Using Aerosol Optical Tweezers and an Electrodynamic Balance. J. Phys. Chem. A 2010, 114 (4), 1806–1815.

Gorkowski, K.; Donahue, N. M.; Sullivan, R. C. Emerging Investigator Series: Determination of Biphasic Core–Shell Droplet Properties Using Aerosol Optical Tweezers. Environ. Sci. Process. Impacts 2018, 20 (11), 1512–1523.

These citations will be added to the relevant portion of the conclusion (Page 22, line 12):

This work also helps inform the design of future experimental work using EDB-MS instrumentation. A measurement of particle diameter with lower uncertainty than the spring point technique would meaningfully reduce the uncertainty in extracted effective vapor pressures, particularly with a continuous diameter measurement. Other research with EDBs has demonstrated the utility of optical sizing techniques for performing this measurement (*Gorkowski et al., 2018; Haddrell et al., 2012; Hargreaves et al., 2010; Marsh et al., 2017; Steimer et al., 2015;* Zardini et al., 2006).

**Miscellaneous updates**

Due to an editing error following two passages were mistakenly omitted from the discussion paper and will be included:

Page 12, line 7:

... where  $X_i$  is the particle-phase mole fraction of species *i*,  $P_{vap_i,eff}$  is the pure component vapor pressure of species *i* at temperature *T* inside the EDB, and *k* is the Boltzmann constant. Because the droplet surface is curved, the vapor pressure of a compound above a charged levitated droplet could conceivably be elevated due to the Kelvin effect. The particle charge also could lower the vapor pressure of a compound with a significant dipole moment. However, following Sect. C1 of Huisman et al. (2013) to calculate the combined Kelvin and charge stabilization effects, we conclude the effects are negligible considering the particle diameter and charge for these experiments.

Page 13, line 24:

The mole fraction of water in the particle,  $X_{H_2O}$ , was assumed to be fixed over the entire course of the experiment, assuming which is true if the activity coefficient of water does not

change appreciably as butenedial evaporates from the particle. We checked the validity of the assumption by using AIOMFAC to compare the change in calculated  $X_{H_2O}$  for the humid, inorganic-free experiment, between its initial composition and composition after all butenedial evaporated. We found the change in calculated  $X_{H_2O}$  to be approximately 0.005, implying the effect of this assumption on our results is negligible within other sources of uncertainty.

The Acknowledgments section has been updated (Page 23, line 1):

This material is based upon work supported by NSF grant CHE 1808084, the National Science Foundation Graduate Research Fellowship under grant numbers DGE 1144152 and DGE 1745303, and the Harvard University Faculty of Arts and Sciences Dean's Competitive Fund for Promising Scholarship. The authors thank Ulrich Krieger *and Steven Wofsy* for useful discussions.